# Vertical Sleeve Gastrectomy Offers Protection against Disturbed Flow-Induced Atherosclerosis in High-Fat Diet-Fed Mice

**DOI:** 10.3390/ijms24065669

**Published:** 2023-03-16

**Authors:** Jih-Hua Wei, Wei-Jei Lee, Jing-Lin Luo, Hsin-Lei Huang, Shen-Chih Wang, Ruey-Hsing Chou, Po-Hsun Huang, Shing-Jong Lin

**Affiliations:** 1Division of Cardiology, Department of Internal Medicine, Min-Sheng General Hospital, Taoyuan 330, Taiwan; jaccwei@yahoo.com.tw (J.-H.W.);; 2School of Medicine, National Defense Medical Center, Taipei 114, Taiwan; 3Institute of Clinical Medicine, National Yang Ming Chiao Tung University, Taipei 112, Taiwan; 4Cardiovascular Research Center, National Yang Ming Chiao Tung University, Taipei 112, Taiwan; 5Department of Surgery, Min-Sheng General Hospital, Taoyuan 330, Taiwan; 6School of Nursing, National Taipei University of Nursing and Health Sciences, Taipei 112, Taiwan; 7Department of Critical Care Medicine, Taipei Veterans General Hospital, Taipei 112, Taiwan; 8Division of Cardiology, Department of Internal Medicine, Taipei Veterans General Hospital, Taipei 112, Taiwan; 9Department of Anesthesiology, Taipei Veteran General Hospital, Taipei 112, Taiwan; 10Department of Medical Research, Taipei Veterans General Hospital, No.201, Sec. 2, Shipai Rd., Beitou District, Taipei 112, Taiwan; 11Taipei Heart Institute, Taipei Medical University, Taipei 110, Taiwan

**Keywords:** bariatric surgery, vertical sleeve gastrectomy, obesity, type 2 diabetes, carotid artery ligation, insulin resistance, atherosclerosis

## Abstract

Bariatric surgery reduces body weight, enhances metabolic and diabetic control, and improves outcomes on obesity-related comorbidities. However, the mechanisms mediating this protection against cardiovascular diseases remain unclear. We investigated the effect of sleeve gastrectomy (SG) on vascular protection in response to shear stress-induced atherosclerosis using an overweighted and carotid artery ligation mouse model. Eight-week-old male wild-type mice (C57BL/6J) were fed a high-fat diet (HFD) for two weeks to induce weight gain and dysmetabolism. SG was performed in HFD-fed mice. Two weeks after the SG procedure, partial carotid-artery ligation was performed to promote disturbed flow-induced atherosclerosis. Compared with the control mice, HFD-fed wild-type mice exhibited increased body weight, total cholesterol level, hemoglobin A1c, and enhanced insulin resistance; SG significantly reversed these adverse effects. As expected, HFD-fed mice exhibited greater neointimal hyperplasia and atherosclerotic plaques than the control group, and the SG procedure attenuated HFD-promoted ligation-induced neointimal hyperplasia and arterial elastin fragmentation. Besides, HFD promoted ligation-induced macrophage infiltration, matrix metalloproteinase-9 expression, upregulation of inflammatory cytokines, and increased vascular endothelial growth factor secretion. SG significantly reduced the above-mentioned effects. Moreover, HFD restriction partially reversed the intimal hyperplasia caused by carotid artery ligation; however, this protective effect was significantly lower than that observed in SG-operated mice. Our study demonstrated that HFD deteriorates shear stress-induced atherosclerosis and SG mitigates vascular remodeling, and this protective effect was not comparable in HFD restriction group. These findings provide a rationale for using bariatric surgery to counter atherosclerosis in morbid obesity.

## 1. Introduction

With the widespread of the western lifestyle, obesity, and in particular extreme obesity, has become a global problem [1]. Importantly, the prevalence of obesity is dramatically increasing in developed and developing countries in the past decades. Approximately one-third of US residents have a body mass index (BMI) exceeding 30; 5% to 10% have a BMI of more than 40 [2,3,4]. Obesity-induced metabolic syndrome is a potent risk factor for atherosclerotic cardiovascular disease and type 2 diabetes mellitus (T2D) [5]. The long-term results of various non-surgical weight-loss interventions are not satisfactory [6,7]. Obesity needs to be well controlled to reduce the risks of associated co-morbidities [8]. Bariatric surgery can effectively treat obesity and resolve obesity-associated co-morbidities [9,10]. Sleeve gastrectomy (SG), the most commonly performed bariatric surgical remedy for the treatment of morbid obesity worldwide [11], reduces the stomach size by ~80% after the removal of a large portion of the greater curvature. Bariatric surgery results in an 80% resolution rate for obesity-associated comorbidities such as T2D [10] and lowered cardiovascular risk [12]. Despite the diversity among studies, bariatric surgery is associated with improving subclinical atherosclerosis and endothelial function, which may meaningfully contribute to reducing the CV risk after a successful weight-reduction operation [13]. Bariatric surgical weight loss is associated with favorable vasculature and reduced cardiovascular mortality [14]. Similar to platelets in atherogenesis, coagulation plays a pivotal role in the histopathogenesis of atherosclerosis, plaque formation, and its stability [15]. However, studies on the potential impact of bariatric surgery on vascular remodeling and intimal hyperplasia in response to vascular injury are limited.

Previously, we showed the efficacy of metabolic surgery for T2D treatment and the reduction of the risk of cardiovascular diseases by up to 50% for 1 year after surgery [16]. Here, we aimed to investigate the influence of SG on disturbed flow-induced atherosclerosis in a mouse model fed with a high-fat diet (HFD) and underwent partial carotid artery ligation. Furthermore, we sought to identify the beneficial effects of SG, compared to diet restriction, in HFD-fed mice in response to shear stress-induced atherosclerosis.

## 2. Results

### 2.1. Effects of SG on Body Weight, Glucose Metabolism, Insulin Resistance, and Cholesterol in HFD-Fed Mice

Hyperglycemia and homeostasis model assessment-estimated insulin resistance (HOMA-IR) were significantly induced (*p* ˂ 0.05) in mice after 2 weeks of HFD consumption (Appendix A), however, insulin levels were not altered significantly (Appendix A). After a 2 week-induction of hyperglycemia, the mice were fed HFD throughout the whole study period. HFD-fed mice showed significantly increased body weight, total cholesterol, and fasting glucose levels at the end of the study compared to mice on a chow diet (Figure 1D–I and Appendix A). We observed significantly elevated glycated hemoglobin (HBA1c) levels and insulin resistance index as assessed by IPGTT and HOMA-IR after HFD administration (Figure 1E through Figure 1H), indicating that HFD induced obesity-related diabetes and insulin resistance. We then performed the SG and sham operations on HFD-fed mice to investigate the impact of SG on insulin resistance and lipid profiles. HFD-fed mice that underwent SG had significantly reduced body weight, total cholesterol, and fasting glucose levels relative to the HFD-fed sham-operated mice (Figure 1D through Figure 1I and Appendix A). Nevertheless, the difference in body weight between mice under restricted HFD and HFD under SG was not significant, which indicated that these metabolic effects were beyond weight reduction. Moreover, SG significantly mitigated insulin resistance as determined by IPGTT, HBA1c, and HOMA-IR in HFD-fed mice.

### 2.2. SG Alleviates HFD-Induced Circulating Inflammatory Cytokines

HFD-fed mice exhibited significantly elevated plasma levels of TNF-α, IL-6, MMP-9, and VEGF compared to mice on a normal chow diet before carotid ligation. SG significantly reduced the elevation of cytokines and MMP-9 levels (Figure 2). Furthermore, the plasma levels of TNF-α, IL-6, MMP-9, and VEGF were significantly upregulated after carotid artery ligation in mice on an HFD. Notably, SG significantly mitigated the plasma levels of TNF-α, IL-6, MMP-9, and VEGF.

### 2.3. Effect of SG on Vascular Remodeling after Carotid Artery Ligation

As shown in Figure 3, carotid artery ligation induced intimal hyperplasia in mice that were fed a regular chow diet—a phenomenon more prominent in HFD-fed mice. HFD-fed mice that underwent SG before carotid ligation exhibited significantly attenuated intimal hyperplasia, indicating that SG offered protection against restricted blood flow-induced vascular remodeling upon HFD feeding (Figure 3A–C). VVG staining showed that HFD promoted elastic fiber proliferation and arterial fragmentation with loss of elastic lamina integrity in response to carotid ligation compared to mice fed with a regular diet. SG significantly attenuated these adverse effects in HFD-fed mice with carotid ligations (Figure 3D,E).

### 2.4. SG Decreased the Inflammatory Expression of CD68, and TNF-α, MMP-9, and VEGF Protein Expression after Carotid Ligation

Carotid artery immunostaining revealed markedly increased macrophage infiltration after HFD feeding (Figure 4A,B). Additionally, SG significantly ameliorated ox-LDL lesion accumulation after carotid ligation in HFD-fed mice (Appendix A). MMP-9 activation was increased after HFD feeding especially after carotid ligation-induced vascular injury (Appendix A). SG significantly mitigated the MMP-9 activation in HFD-fed mice. Moreover, the protein levels of TNF-α, MMP-9, and VEGF were markedly activated after carotid artery ligation in mice on an HFD, which were ameliorated by SG in HFD-fed mice with carotid ligation (Figure 4C through Figure 4F).

### 2.5. HFD Restriction Had Less Protective Effect on Carotid-Ligated Vascular Remodeling Than SG Surgery

Mice on a restrictive HFD (30% caloric reduction of HFD) showed a similar body weight reduction as SG-operated HFD-fed mice (Appendix A). HFD induced prominent intimal hyperplasia and elastic fiber proliferation with the loss of elastin integrity after carotid ligation, whereas SG and HFD restriction significantly reduced these effects. However, mice with food restrictions exhibited more prominent intimal hyperplasia and elastic fiber proliferation, compared to SG-operated HFD-fed mice (Appendix A). Mice on a restricted HFD with carotid ligation had fewer effects on reducing ox-LDL, CD68, and MMP-9 accumulation (Appendix A). We observed macrophage infiltration and MMM-9 upregulation compared to SG-operated HFD-fed mice indicated vascular repair enhancement after injury.

## 3. Discussion

In this study, we induced weight gain and obesity-associated Mets, including insulin resistance and enhanced local and systemic inflammation, in wild-type mice by administering an HFD. Consistent with previous studies, elevated body weight and high circulating glucose levels were observed in a wild-type mouse after intake of HFD [17]. Insulin levels increased progressively with time, and the mice exhibited insulin resistance and glucose intolerance under intravenous glucose challenge [17]. The fasting glucose, insulin level, and HOMA-IR were all consistently elevated when the assigned HFD continued beyond the eight weeks [17,18]. Moreover, HFD-fed mice exhibited more significant neointimal hyperplasia and atherosclerotic plaques than the control group. The SG procedure significantly attenuated HFD-promoted ligation-induced neointimal hyperplasia and arterial elastin fragmentation. Interestingly, HFD restriction partially reversed the intimal hyperplasia caused by carotid artery ligation, and this protective effect was lower than that observed in SG-operated mice. These findings provide a rationale for using bariatric surgery to counter atherosclerosis in morbid obesity possibly through the reversal of insulin resistance, leading to the amelioration of neointimal hyperplasia.

The cytokines IL-6, TNF-α, and leptin play critical roles in the development of obesity, insulin resistance, and chronic inflammation, which culminate into obesity-related metS [19]. Weight loss after laparoscopic SG is associated with a considerable reduction of body weight and adipokine levels tend toward anti-diabetic and anti-inflammatory profiles [20,21]. We showed that HFD and carotid ligation increased TNF-α, IL-6, MMP-9, and VEGF levels in circulation, indicating an induction of systemic inflammation. Plasma levels of TNF-α, IL-6, MMP-9, and VEGF were significantly upregulated after carotid artery ligation in mice on a normal chow diet and HFD. Obesity-associated inflammation plays a causative role in generating insulin resistance [22,23]. A study in type II diabetic rats showed that insulin resistance is associated with balloon injury-related neointimal hyperplasia [24]. In this study, HFD-enhanced TNF-α, ox-LDL, IL-6, and MMP-9 levels, whereas SG markedly downregulated proinflammatory cytokines and reversed insulin resistance, resulting in vascular protection. This study has shown that insulin resistance is linked to intimal hyperplasia and atherosclerosis. Additionally, SG reverses HFD-induced obesity and reduces vascular inflammation. Moreover, this effect was more pronounced than that induced by dietary restriction.

Previous studies had demonstrated lower intimal hyperplasia due to wire-induced carotid artery injury in TNF-α-deficient TNF (-/-) mice. However, TNF-α and NF-κB expression was enhanced in wild-type mice subjected to wire carotid artery injury [25]. Rectenwald et al. demonstrated that TNF-α and IL-1 modulate low shear stress–induced neointimal hyperplasia. This study shows that proinflammatory cytokine signaling directly links biomechanical forces on the vessel wall and vascular remodeling [26]. The mechanisms underlying obesity-associated inflammation remain elusive, and the precise triggers may differ between tissues [27]. The innate immune system is activated in obesity; M1-polarized macrophages that display a pro-inflammatory phenotype and secrete cytokines such as TNF-α are increased during obesity [28]. IL-6, which is synthesized and released by macrophages, smooth muscle cells, and endothelial cells, plays a pivotal role in the early stages of atherosclerotic plaque formation. IL-6 can be resident or trapped in arterial walls, playing an important role in the phenotypic determination of plaque macrophages, polarization and shaping of macrophages, and development of vascular intimal hyperplasia and atherosclerosis [29]. Although we did not examine macrophage composition, studies have indicated that M2 macrophages are polarized after SG, which may be relevant to the results of vascular intimal hyperplasia and inflammation [30,31]. Here, we demonstrated the SG improves TNF-α expression, macrophage infiltration, and IL-6 level altered by carotid ligation in HFD-fed mice, eventually resulting in amelioration of intimal hyperplasia.

For tissue inflammation analysis, we focused on vascular remodeling after injury. SG downregulated VEGF, which is a crucial mediator for neointimal formation and progression [32]; its expression was induced by HFD feeding. SG and carotid artery ligation are surgical interventions for weight reduction and vascular remodeling after injury, respectively. In line with previous studies, a HFD augmented inflammatory responses toward vascular injury with increased macrophage infiltration, MMP-9 expression, upregulation of inflammatory cytokines, and VEGF secretion [33,34,35,36]. SG after providing an HFD significantly attenuated vascular injury-induced inflammatory responses and consequent vascular remodeling that had a minor degree of neointimal hyperplasia and arterial elastin fragmentation. However, diet restriction ameliorates obesity-associated inflammation and body weight loss [37]. In addition to weight loss, SG provided additional benefits during vascular remodeling after injury compared to HFD restriction by improving ox-LDL and MMP-9 expression and macrophage infiltration. Our results suggest a potential therapeutic rationale for the use of bariatric surgery to treat obesity and associated comorbidities.

However, our results indicated that diet-restricted mice exhibited severe neointimal hyperplasia and increased rupture of collagen integrity compared to mice that underwent SG. SG performed before carotid ligation attenuated adverse vascular remodeling in response to shear stress-induced atherosclerosis. Studies indicated that the stress-responsive hypothalamic-pituitary-adrenocortical axis may play a role in the diverse protective effects conferred by SG and diet restriction [38,39]. Here, we demonstrated the protective effects of SG against anti-obesity-associated-inflammation and vascular remodeling under shear stress via histology in comparison to diet restriction.

Increased ox-LDL accumulation, macrophage infiltration, IL-6 increasing and MMP-9 activation are associated with unstable plaque rupture and enhanced vascular remodeling in patients with CAD [40,41,42,43]. Hinagata et al. demonstrated that the expression of the ox-LDL receptor LOX-1 in smooth muscle cells is involved in intimal hyperplasia during balloon injury [44]. Our results demonstrated that SG attenuated macrophage infiltration, ox-LDL accumulation, IL-6 increasing and MMP-9 expression, indicating that SG reduced oxidative stress, macrophage bioactivity, and vascular inflammation, and may cause plaque stabilization and decrease cardiovascular events. Gokce et al. demonstrated that bariatric intervention improves vascular outcomes by evaluating flow-mediated dilation and reactive hyperemia in a wide range of individuals with cardiovascular risk [14]. For obesity control, both non-surgical and surgical interventions confer beneficial effects [45,46]. Studies have shown the positive impact of bariatric surgery on albuminuria, endothelial function, and inflammation in patients with obesity and type 2 diabetes [47,48]. Further investigation is needed to elucidate whether these protective effects mitigate obesity-associated cardiovascular risks.

Our original study design did not directly incorporate a comparison of the outcomes between SG and diet restriction. However, our results indicated additional beneficial effects of SG in vascular remodeling. Nevertheless, the differences between the underlying protective mechanisms of SG and diet restriction and their clinical impact on cardiovascular risks warrant further investigations.

Obesity is associated with prothrombotic status and increased risk of thrombotic events. Bariatric surgery reduces body weight, inflammation, and the activation of extrinsic coagulation pathways. The reduction in protein C (PC), activated PC, soluble thrombomodulin (TM), and soluble E-selectin levels a year after Roux-en-Y gastric bypass surgery suggests a compensatory upregulation of PC during obesity. The reduction in TM and E-selectin suggests improved endothelial function in this cohort of patients [49]. TM deficiency in endothelial cells resulted in increased basal permeability and hyperpermeability when stimulated using thrombin and TNF-α. These results suggest that cell-bound TM maintains a quiescent phenotype in vascular endothelial cells by regulating the expression of procoagulant and proinflammatory molecules [50]. Proinflammatory cytokines may precipitate thrombus formation, activate the endothelial coagulation system, and promote the development of acute coronary syndromes [51]. Unfortunately, this issue was not explored in the current study but will be our main focus in future studies on the link between weight-reduction surgery and intimal hyperplasia-associated atherosclerosis.

## 4. Methods

### 4.1. Experimental Animals

Wild-type (WT) male C57BL/6J mice (total 105 mice; age, 8-week-old; average weight, 16–18 g) were purchased from the National Laboratory Animal Center (NLAC). Mice were housed at the National Yang Ming Chiao Tung University Laboratory Animal Center with a 12–12 h light-dark cycle (light onset at 7:00 A.M). The mice were provided standard rodent chow and water ad libitum. Procedural approval for animal experiments was obtained from the National Yang Ming Chiao Tung University Institutional Animal Care and Use Committee. Our study complied with the “Guide for the Care and Use of Laboratory Animals,” 8th edition, 2011. The animals were subjected to restricted feeding with free access to water on the night before surgery.

### 4.2. Animal Grouping and Diets

Mice were randomly assigned to six groups (n = 15 each group). Group 1 (CSS group) mice were fed chow, and were subjected to sham operations for SG and carotid ligation surgery. Group 2 (CSL) mice were fed chow and underwent carotid ligation alone. Group 3 (HSS) mice were provided HFD and were sham-operated for SG and carotid ligation. Group 4 (HSL) mice were fed HFD and were subjected to sham surgery for SG but received carotid ligation. Group 5 (HGS) mice were fed HFD and underwent SG and sham operation for carotid ligation. Group 6 (HGL) mice were fed HFD and underwent SG and carotid ligation surgeries. The timeline for the procedures, blood sampling, and tissue processing is shown in Figure 1A. Chow diet and HFD were continued throughout the whole study period. We also assessed the effect of diet restriction without sleeve gastrectomy in mice. Mice under strict diet restriction were fed 70% HFD mixed with 30% wheat bran. Chow diet comprised of Picolab Rodent Diet 20 (energy, 4.06 kcal/g; protein, 23.5%; fat, 11.3%; carbohydrates, 50.3% (13.5% energy by fat). High fat diet comprised of Teklad diet TD 88137 (energy, 4.55 kcal/g; protein, 17.8%; fat, 20.2%; carbohydrates, 50.5% (39.9% energy by fat) Harlan Tackle Co. Body weight of mice was assessed weekly in all experiments.

### 4.3. SG

SG was performed using the clip applier technique [52]. Anesthetic and aseptic procedures, antibiotic indication and usage, and post-operative pain control were performed as previously described [53]. Mice underwent fasting ~6 h before surgery. The hair over the upper abdomen was removed and a 1–1.5 cm upper midline incision was made. SG was performed using a magnifying dissecting microscope (MICROTEK SZ5T-ST^®^, Telescopes, Taipei, Taiwan) and magnifier as needed to prevent unpredictable blood loss with eventual micro suture closure. After the stomach was externalized, the gastric fundus from the surrounding tissue and other internal organs were dissected. As the fundus and pylorus of the stomach were stretched gently and laterally with micro-forceps, the midline was identified as in Figure 1B; the Ligaclip Applier (Ethicon Inc., Somerville, NJ, USA) was carefully applied to the 80% medial side of the stomach inferiorly from the gastroesophageal junction and superiorly from the lower pole. Approximately 80% of the stomach was clamped and excluded; the stomach’s entire lateral sleeve was created. The stomach’s excluded sleeve was removed, sterilized, and the clipped line was over-sewn with a 6-0 nonabsorbable monofilament suture to ensure no leakage. The stomach was returned to the original site in the abdominal cavity; the abdomen was closed by running a 5-0 nonabsorbable monofilament discontinuous suture to the fascia and abdominal wall layers separately. For the SG sham procedure, the mice received the same laparotomy, externalizing of the stomach using a wet warm saline gauze coverage for 5 min. The stomach was returned to the abdominal cavity; the abdominal wall was closed as described previously.

### 4.4. Partial Ligation of the Left Carotid Artery

To establish a model of disturbed flow-induced vascular remodeling in vivo, we partially ligated the left carotid arteries (LCAs) in mice as described previously [54,55,56,57]. The hair over the upper chest and neck was removed. Povidone-iodine solution was used to follow aseptic procedures and a 4–6 mm midline vertical incision was made. The soft tissue was dissected laterally, and the left common carotid artery was identified using the above-mentioned magnifying dissecting microscope. The external carotid artery, internal carotid artery, occipital artery, and superior thyroid artery were carefully dissected from the carotid bulb. The external carotid artery above the superior thyroid artery was tied off using a 6-0 silk suture. The internal carotid artery was similarly tied off contemporaneously. The skin was approximated and closed using a 5-0 monofilament nonabsorbable suture. For the sham partial carotid ligation, the skin and fascia over the upper chest wall was opened; the LCA and associated tissues were identified as described above. The tissue was returned to the original sites and the skin and fascia were closed.

### 4.5. Morphometric Analysis

Four weeks after carotid artery ligation, the mice were euthanized using an intraperitoneal 250 mg/kg avertin injection. The left ventricle was cannulated, perfused with phosphate-buffered saline (PBS), and fixed with 4% paraformaldehyde (PFA) (Sigma-Aldrich, St. Louis, MO, USA). The left and right carotid arteries were collected and incubated in 4% PFA for 8 h. After cryopreservation in 30% sucrose/PBS at 4 °C, the arteries were embedded in the Tissue-Tek optimal cutting temperature compound and frozen. Cross-sections of 5 µm thickness were obtained ~1 mm proximal to the bifurcation of the common carotid artery and stained with hematoxylin and eosin (H&E). Four regions (the lumen, intima, media, and total vascular area) of the H&E-stained cross-sections were examined using ImageJ software (National Institutes of Health). The areas surrounding the luminal surface, internal elastic lamina, and external elastic lamina were measured. The intimal area was determined by subtracting the luminal area from the area defined by the internal elastic lamina. The medial area was calculated by subtracting the area defined by the internal elastic lamina from the external elastic lamina.

### 4.6. Elastic Fragmentation Measurement

Frozen carotid artery sections were harvested and stained with Verhoeff-van Gieson (VVG; Sigma-Aldrich) stain to distinguish the elastic fibers. The sections were hydrated and stained in Verhoeff’s solution for 10 min. Next, the sections were incubated in 2% ferric chloride for 2 min and treated with 5% sodium thiosulfate for 1 min. The sections were counterstained in VVG for 5 min and dehydrated. Fragmentation was defined as the presence of noticeable cracks in the continuous elastin fiber. Images were acquired using the Olympus BX63 microscope (Olympus, Center Valley, PA, USA); the number of elastin fragments was estimated in representative VVG images for each mouse carotid artery.

### 4.7. Postoperative Care

The mice were placed on a warm pad after the operation or sham procedure; an oxygen flow of 2 L/min was given until the mice were fully awake. Post-operative mice were under surveillance in an isolated chamber as they regained mobility and resumed walking around the cage. A single dose of meloxicam (0.1 mg/kg) was administered intraperitoneally for pain relief after surgery and once daily as required for 3 days. An intraperitoneal injection of cefazolin (25 mg/kg) was administered for 1 day after the operation. The mice were kept in an independent incubator at 30 °C for 5 days. One mouse was housed per cage to prevent injury. The mice were free-fed a high-fat gel diet (10% lard, 10% liquid sugar, 57% water) for 3 days after surgery and were subsequently reintroduced to the previous diet. The mouse body weight was measured weekly throughout the study period. Blood samples and tissue specimens were harvested as scheduled. There was no major bleeding or other significant life-threatening side effect noted during the experimental surgeries. The mortality rate was less than 5% at the early stage of the study. In our hands, the success rate of partial carotid ligation surgery and sleeve gastrectomy was 100% and higher than 90%, respectively.

### 4.8. Intraperitoneal Glucose Tolerance Test (IPGTT)

After overnight fasting of mice (for at least 10 h), a baseline blood sample was taken (0 min) before the intraperitoneal glucose injection (1 mg/g of body weight). The blood glucose levels were measured at 5, 15, 30, 60, and 120 min after glucose challenge using ACCU-CHEK glucometers (Roche, Basel, Switzerland) and test strips. All blood samples were obtained from the tail vein of freely moving mice using topical cream anesthesia to cause less pain and panic [58,59].

### 4.9. Histology and Immunohistochemistry

The 2–3 mm long common carotid arterial tissues were obtained 1 mm near the carotid bifurcation and were fixed for pathologic evaluation using a 4% PFA solution. Cross-sections of 5 µm were obtained 1 mm proximal to the bifurcation and were stained with H&E and VVG for elastin. Duplicate frozen sections were used for immunohistochemical staining. Briefly, sections were incubated with primary antibodies against oxidized low-density lipoprotein (ox-LDL, 1:500, TA336722, OriGene Technologies, Rockville, MD, USA), CD68 (1:10, NB100-683, Novus-Biologicals, Littleton, CO, USA), and MMP-9 (1:500, SAB5200294, Sigma-Aldrich). The sections were incubated with a biotinylated secondary antibody. An avidin-biotin complex kit (Cell and Tissue Staining Kit, R&D System, Minneapolis, MN 55413, USA) was used for detection.

### 4.10. Measurement of Inflammatory Cytokines

Blood samples were collected from each mouse and centrifuged at 3000× *g* for 10 min at 4 °C. The plasma was transferred to separate tubes without disturbing the blood clots and stored at −80 °C. Plasma samples were analyzed using mouse ELISA kits (R&D System, Minneapolis, MN, USA) for vascular endothelial growth factor (VEGF), pro-matrix metalloproteinase 9 (MMP-9), IL-6, and tumor necrosis factor-α (TNF-α) as per the manufacturer’s instructions.

### 4.11. Western Blotting

The carotid arteries were harvested and homogenized in radioimmunoprecipitation assay (RIPA) buffer (R0278; Sigma-Aldrich). After centrifugation at 4 °C, the supernatants were collected. Protein concentration was determined using a Bio-Rad Protein Assay Kit (#5000006; Bio-Rad Laboratories, Hercules, CA, USA). Briefly, 20 µL of 2× SDS sample buffer was added to each sample and heated for 10 min. The samples were loaded onto a 12% Tris-HCl gel (Bio-Rad Laboratories, Hercules, CA, USA) and run at a constant voltage (80 V). Proteins were then transferred to a polyvinylidene fluoride membrane (Bio-Rad Laboratories) for blotting using a constant 350 mA current. Blocking was performed for 1 h with 5% bovine serum albumin (BSA) in phosphate-buffered saline with 0.2% Tween-20 (PBS-T). Membranes were incubated overnight at 4°C with mouse antibody against MMP-9 (1:500, sab5200294, Sigma-Aldrich), mouse monoclonal anti-β-Actin antibody (1:20,000), goat anti-mouse VEGF (1:500, V1253, R&D System), and mouse monoclonal TNF-α antibody (1:2000, T0938, Sigma-Aldrich). After incubation, the blots were washed with PBS-T and were incubated with horseradish peroxidase-conjugated mouse anti-rabbit and mouse antibodies (Dako, Glostrup, Denmark) (1:1000, in PBS-T) at room temperature for 1 h. The enhanced chemiluminescence system (ECL Western Blotting Detection Reagents, Millipore Corporation, USA) and Luminescence/Fluorescence Imaging System (LAS4000, Fujifilm) were used for detection.

### 4.12. Statistical Analysis

Data are expressed as mean ± standard deviation. Multiple groups comparisons were performed using one-way analysis of variance followed by Scheffe’s and Tukey multiple-comparison post hoc test. Paired-t tests were used to analyze the statistical significance of the effects of treatments. Two-way analysis of variance was also performed with one treatment parameter being normal chow diet, HFD, HFD with SG and carotid ligation surgery/no surgery. A *p*-value < 0.05 in the two-way ANOVA tests were considered statistically significant. *p*-values for the trend and one-way ANOVA were estimated for diet intake and serial changes of glucose in IPGTT. Analyses were conducted using SPSS software (version 22; SPSS, Chicago, IL, USA). *p*-values < 0.05 were considered statistically significant.

## 5. Conclusions

Our findings demonstrated that HFD deteriorates shear stress-induced atherosclerosis and SG significantly attenuated vascular remodeling (graphical abstract). HFD restriction partially reversed the intimal hyperplasia caused by carotid artery ligation, and this vasoprotective effect was lower than that observed in SG-operated mice. These findings provide a rationale for using bariatric surgery to counter atherosclerosis in morbid obesity.

## Figures and Tables

**Figure 1 ijms-24-05669-f001:**
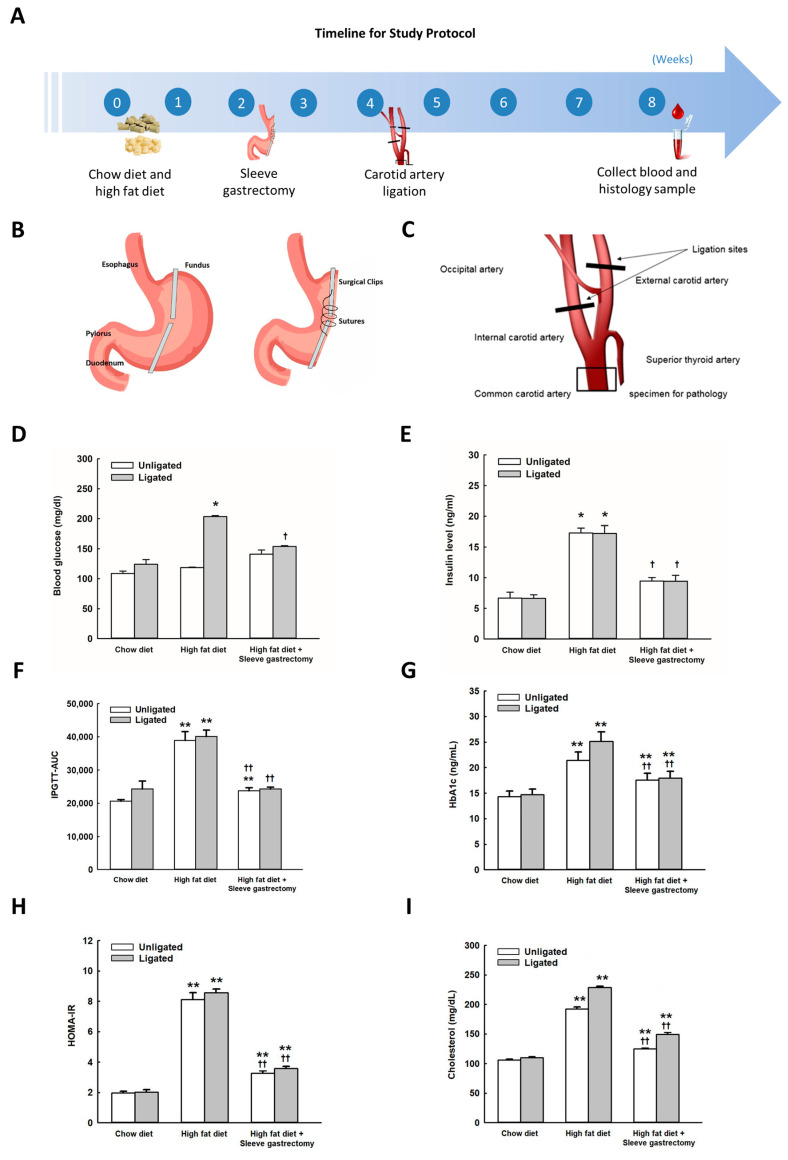
Effects of SG on glucose and cholesterol metabolism in HFD-fed mice. Timeline for the experimental procedures, blood sampling, and tissue processing during the study (**A**). The SG and carotid artery ligation procedures are illustrated in (**B**,**C**). Changes of glucose, insulin, and homeostatic assessment model for insulin resistance (HOMI-IR) were assessed (**D**–**H**). The cholesterol level in mice increased significantly after HFD and decreased significantly after SG (**I**). The intraperitoneal glucose tolerance test and area under curve (IPGTT-AUC), hemoglobin A1c (HbA1c), HOMA-IR for glucose tolerance and insulin resistance survey were measured (**G**–**I**) (n = 12–15 per group). The *p* for the trend of IPGTT with one-way ANOVA and Scheffee post-hoc test was all significant with a *p* = 0.003. The differences between chow diet, HFD, and HFD with SG were significant. Significant differences were not observed in the above-mentioned groups that did/did not undergo carotid artery ligation. The difference in IPGTT in HFD-fed mice that underwent SG was significant compared with the HFD-fed mice without SG. A significant difference was not observed upon comparison with control chow-fed mice. Chow diet: Picolab Rodent Diet 20; 4 kcal/g, 2% cholesterol, 5.7% fat; High fat diet: Teklad diet TD 88137; 4.5 kcal/g, 0.2% cholesterol, 21.2% fat; Harlan Tackle Co. * High-fat diet (HFD) compared with chow diet (CD) in mice that underwent carotid artery ligation or those that did not. * *p* < 0.05, ** *p* < 0.005, ^†^ Mice that underwent SG compared to those that did not undergo SG with or without carotid ligation. ^†^ *p* < 0.05, ^††^ *p* < 0.005. The *p*-value for trend and one-way ANOVA were estimated for diet intake and serial changes of glucose in IPGTT with Scheffee and Tukey post hoc tests.

**Figure 2 ijms-24-05669-f002:**
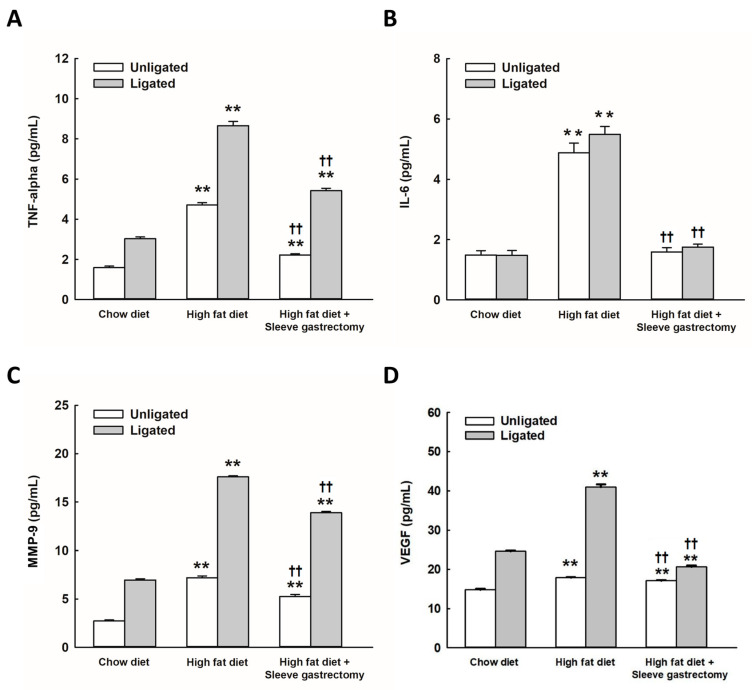
SG alleviated the levels of circulating HFD-induced inflammatory cytokines. The circulating concentrations of TNF-α (**A**), IL-6 (**B**), MMP-9 (**C**), and VEGF (**D**) as estimated by ELISA in mice. (n = 12 per group), Bar graphs indicate mean ± SD; High-fat diet (HFD) compared with chow diet (CD) in mice with or without carotid ligation, ** *p* < 0.005, ^††^ *p* < 0.005. Two-way analysis of variance was performed with one treatment parameter being normal chow, HFD, and HFD with SG, and the other parameter being carotid ligation/no carotid ligation. A *p*-value < 0.05 was considered to be statistically significant.

**Figure 3 ijms-24-05669-f003:**
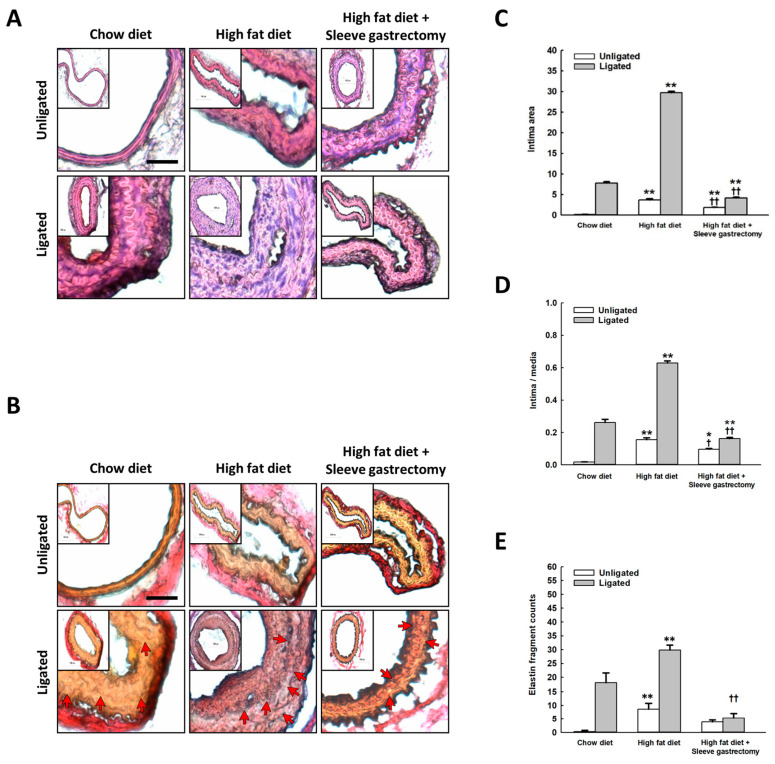
SG attenuated HFD-induced intimal thickening. Representative light micrographs from the ligated left common carotid artery of the mice. H&E and VVG staining were performed (**A**,**B**) Intima, media areas, intima/media area ratio, and arterial fragmentation counts in all groups were measured. Two-way analysis of variance was performed with one treatment parameter being normal chow, HFD, and HFD with SG, and the other parameter being carotid ligation/no carotid ligation. A *p*-value < 0.05 was considered to be statistically significant. (**C**–**E**) (n = 6 per group), Scale bars = 100 μm. * High-fat diet (HFD) compared with chow diet (CD) in mice with or without carotid ligation, * *p* < 0.05, ** *p* < 0.005, ^††^ *p* < 0.005.

**Figure 4 ijms-24-05669-f004:**
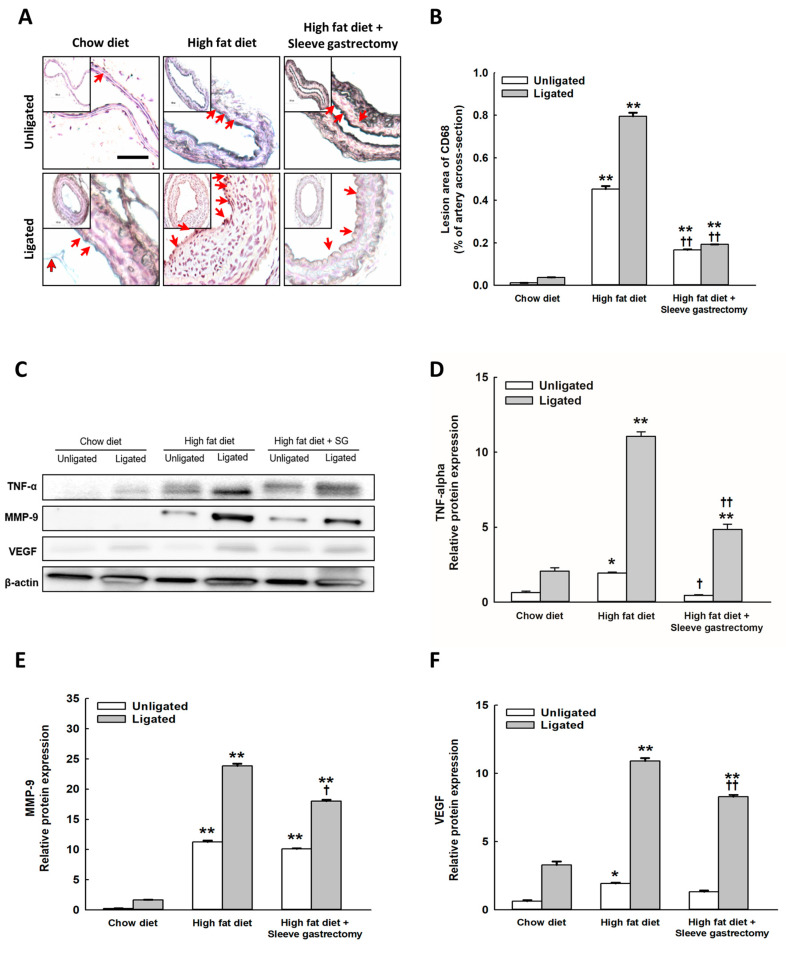
SG ameliorated HFD induced inflammatory reaction and protein expression in the ligated carotid artery. Representative light micrographs from the ligated left common carotid artery of the mice. Carotid artery immunostaining revealed ox-LDL lesions, macrophage infiltration, and MMP-9 expression (**A**,**B** and Appendix A through Appendix A). The right panels are the quantified immunohistochemical results by densitometry analysis (**B** and Appendix A). Scale bars: 100 μm. (n = 6 per group). The proteins expression of TNF-α, MMP-9, and VEGF were determined in the carotid artery (**C**–**F**). Data are illustrated as the mean ± SD. (n = 4 per group), * High-fat diet (HFD) compared with chow diet (CD) in mice with or without carotid ligation, Two-way analysis of variance was also performed with one treatment parameter being normal chow, HFD, and HFD with SG, and the other parameter being carotid ligation/no carotid ligation. A *p*-value < 0.05 was considered to be statistically significant. * *p* < 0.05, ** *p* < 0.005, ^†^ Mice with SG with or without carotid ligation, ^†^ *p* < 0.05, ^††^ *p* < 0.005.

## Data Availability

The data relevant to the current study are available from the corresponding author upon reasonable reasons.

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
