# Peer review of "Vertical Sleeve Gastrectomy Offers Protection against Disturbed Flow-Induced Atherosclerosis in High-Fat Diet-Fed Mice"

_ijms, 2023, doi:10.3390/ijms24065669_

Round 1

Reviewer 1 Report

In this fascinating study Wei er al. demonstrated that high-fat- diet (HFD) is responsible for stress induced atherosclerosis and sleeve gastrectomy (SG) ameliorates gastric remodeling, reverses intimal-hyperplasia caused by carotid ligation and reduces inflammation. This experimental study is well conducted and presented. It offers food for thoughts about the potential direct effect of bariatric surgery on cardiovascular risks. In my opinion, some minor issues should be addressed to improve definitely the study. Below are some comments and suggestions:

1. It would have been very interesting to study the comparison of main outcomes between a diet-restriction cohort and SG.

2. I think the paragraphs about reduced inflammatory expression of CD68, and TNF-α, MMP-9, and VEGF protein is one of the 
most appealing part of the results and deserve more space in discussion and should be also reported in the conclusion. Reduction of inflammatory markers could be the basis of 
reduction of intimal hyperplasia and could represents the corner point for the possible reduction of atherosclerosis and cardiovascular risk. Please provide insights about inflammation coagulation, intimal media hyperplasia and risk of atherosclerosis. I suggest some interesting studies:

Miceli G, Basso MG, Rizzo G, Pintus C, Tuttolomondo A. The Role of the Coagulation System in Peripheral Arterial Disease: Interactions with the Arterial Wall and Its Vascular 
Microenvironment and Implications for Rational Therapies. Int J Mol Sci. 2022 Nov 29;23(23):14914. doi: 10.3390/ijms232314914. PMID: 36499242; PMCID: PMC9739112.

-Blum A, Tamir S, Hazzan D, Podvitzky O, Sirchan R, Keinan-Boker L, Ben-Shushan RS, Blum N, Suliman LS, Geron N. Gender effect on vascular inflammation following bariatric surgery. Eur Cytokine Netw. 2012 Oct-Dec;23(4):154-7. doi: 10.1684/ecn.2012.0318. PMID: 23306174

- Gokce N, Karki S, Dobyns A, Zizza E, Sroczynski E, Palmisano JN, Mazzotta C, Hamburg NM, Pernar LI, Carmine B, Carter CO, LaValley M, Hess DT, Apovian CM, Farb MG. Association of Bariatric Surgery With Vascular Outcomes. JAMA Netw Open. 2021 Jul 1;4(7):e2115267. doi: 10.1001/jamanetworkopen.2021.15267. PMID: 34251443; PMCID: PMC8276087;

-Rega-Kaun G, Kaun C, Ebenbauer B, Jaegersberger G, Prager M, Wojta J, Hohensinner PJ. Bariatric surgery in morbidly obese individuals affects plasma levels of protein C and thrombomodulin. J Thromb Thrombolysis. 2019 Jan;47(1):51-56. doi: 10.1007/s11239-018-1744-9. PMID: 30259314; PMCID: PMC6336753.]

Author Response

Thank you very much for your detail comments. These comments are very instructive and very helpful to this manuscript and our future research. We have tried our best to reply to your comments point-by-point and revise the manuscript accordingly. The responses to your comments are dictated below.

Reviewer(s)' Comments to Author:

Reviewer: 1

Comments to the Author

In this fascinating study Wei et al. demonstrated that high-fat- diet (HFD) is responsible for stress induced atherosclerosis and sleeve gastrectomy (SG) ameliorates gastric remodeling, reverses intimal-hyperplasia caused by carotid ligation and reduces inflammation. This experimental study is well conducted and presented. It offers food for thoughts about the potential direct effect of bariatric surgery on cardiovascular risks. In my opinion, some minor issues should be addressed to improve definitely the study. Below are some comments and suggestions:

  1. It would have been very interesting to study the comparison of main outcomes between a diet-restriction cohort and

Reply: Thank you for your comment. The results of this study reveal that the effects of SG on vascular protection was greater than that observed for diet restriction. As suggested, we have mentioned this interesting fact in lines 262–266 of the Discussion section.

“ Our original study design did not directly incorporate a comparison of the outcomes between SG and diet restriction. However, our results indicated additional beneficial effects of SG in vascular remodeling. Nevertheless, the differences between the underlying protective mechanisms of SG and diet restriction and their clinical impact on cardiovascular risks warrant further investigations.” It would be the target in the future study. Thank you again.

  1. I think the paragraphs about reduced inflammatory expression of CD68, and TNF-α, MMP-9, and VEGF protein is one of the most appealing parts of the results and deserve more space in discussion and should be also reported in the conclusion. Reduction of inflammatory markers could be the basis of reduction of intimal hyperplasia and could represents the corner point for the possible reduction of atherosclerosis and cardiovascular risk. Please provide insights about inflammation coagulation, intimal media hyperplasia and risk of atherosclerosis. I suggest some interesting studies:

Reply: Thank you for your valuable comment. Based on your suggestions, we have incorporated major modifications in the discussion to improve clarity and reasoning. The important and relevant articles that you had suggested have been read and cited in the Discussion section.

- Miceli G, Basso MG, Rizzo G, Pintus C, Tuttolomondo A. The Role of the Coagulation System in Peripheral Arterial Disease: Interactions with the Arterial Wall and Its Vascular 
Microenvironment and Implications for Rational Therapies. Int J Mol Sci. 2022 Nov 29;23(23):14914. doi: 10.3390/ijms232314914. PMID: 36499242; PMCID: PMC9739112.

-Blum A, Tamir S, Hazzan D, Podvitzky O, Sirchan R, Keinan-Boker L, Ben-Shushan RS, Blum N, Suliman LS, Geron N. Gender effect on vascular inflammation following bariatric surgery. Eur Cytokine Netw. 2012 Oct-Dec;23(4):154-7. doi: 10.1684/ecn.2012.0318. PMID: 23306174

- Gokce N, Karki S, Dobyns A, Zizza E, Sroczynski E, Palmisano JN, Mazzotta C, Hamburg NM, Pernar LI, Carmine B, Carter CO, LaValley M, Hess DT, Apovian CM, Farb MG. Association of Bariatric Surgery With Vascular Outcomes. JAMA Netw Open. 2021 Jul 1;4(7):e2115267. doi: 10.1001/jamanetworkopen.2021.15267. PMID: 34251443; PMCID: PMC8276087;

-Rega-Kaun G, Kaun C, Ebenbauer B, Jaegersberger G, Prager M, Wojta J, Hohensinner PJ. Bariatric surgery in morbidly obese individuals affects plasma levels of protein C and thrombomodulin. J Thromb Thrombolysis. 2019 Jan;47(1):51-56. doi: 10.1007/s11239-018-1744-9. PMID: 30259314; PMCID: PMC6336753.]

Thank you very much again. We appreciate your comments!

Reviewer 2 Report

1. Figures are missing. Figure 1-5 mentioned in the article can't be seen. Only supplementary documents

2. Some of the content of the supplementary pictures mentioned in the article is inconsistent with the actual pictures. For example, as mentioned in 3.4 SG significantly ameliorated ox-LDL lesion accumulation after carotid ligation in HFD-fed mice (Supplementary Figure 2A and B). MMP-9 activation was increased after HFD feeding especially after carotid ligation-induced vascular injury (supplementary figure 2C and D). Please check the article carefully.

3. It is necessary to briefly describe the relationship between obesity, intimal hyperplasia and atherosclerosis, insulin resistance, inflammation, etc

4. It is suggested to describe the sample information in 2.11 in detail.

5. In Suppiementary Figure 1B, the unit of mouse body weight is wrong.

6. The article does not mention supplementary figure 4.

Author Response

Thank you very much for your detail comments. These comments are very instructive and very helpful to this manuscript and our future research. We have tried our best to reply to your comments point-by-point and revise the manuscript accordingly. The responses to your comments are dictated below.

Reviewer(s)' Comments to Author:

Reviewer: 2

Comments to the Author

  1. Figures are missing. Figure 1-5 mentioned in the article can't be seen. Only supplementary documents.

Reply: We apologize for the missing data. We have re-upload the figures. Please do check them. Thank you for bringing this to our attention.

  1. Some of the content of the supplementary pictures mentioned in the article is inconsistent with the actual pictures. For example, as mentioned in 4“SG significantly ameliorated ox-LDL lesion accumulation after carotid ligation in HFD-fed mice (Supplementary Figure 2A and B). MMP-9 activation was increased after HFD feeding especially after carotid ligation-induced vascular injury (supplementary figure 2C and D).” Please check the article carefully.

Reply: Thank you for bringing this to our attention; we apologize for the error. We have revised the sequence of the supplementary figures, as suggested. “Under immunohistochemical staining, we found that HFD significantly increased ox-LDL and MMP-9 expression. After carotid ligation, the expression was further enhanced, and sleeve gastrectomy significantly ameliorated ox-LDL accumulation and MMP-9 activation.”

  1. It is necessary to briefly describe the relationship between obesity, intimal hyperplasia and atherosclerosis, insulin resistance, inflammation, etc.

Reply: Thank you for this suggestion. Accordingly, we have identified the defects in our current discussion and have made extensive modifications and detailed additions to the discussion regarding the relationship between obesity, NIH, and atherosclerosis.

  1. It is suggested to describe the sample information in 11 in detail.

Reply: Thank you for your comment. We have provided the sample information and changed to the Section 4.11 according to the journal format.

  1. In Supplementary Figure 1B, the unit of mouse body weight is wrong.

Reply: We apologize for the error. We have corrected it. Thank you once again for bringing this to our notice.

  1. The article does not mention supplementary figure 4.

Reply: We apologize for the error. We have rectified this issue.

Thank you very much again. We appreciate your comments!

Round 2

Reviewer 2 Report

Through modification and supplement, I think it can be accepted